# Divergent Gene Expression Profiles in Alaskan Sea Otters: An Indicator of Chronic Domoic Acid Exposure?

**Lizabeth Bowen [1,***, Susan Knowles [2], Kathi Lefebvre [3], Michelle St. Martin [4,5], Michael Murray [6], Kim Kloecker [7], Daniel Monson [7], Benjamin Weitzman [4], Brenda Ballachey [7], Heather Coletti [8], Shannon Waters [1] and Caroline Cummings [4]**

1. U.S. Geological Survey, Davis, CA 95616, USA
2. U.S. Geological Survey, Madison, WI 53711, USA
3. Environmental and Fisheries Sciences Division, Northwest Fisheries Science Center, National Marine Fisheries Service, NOAA, Seattle, WA 98112 USA
4. U.S. Fish and Wildlife Service, Marine Mammals Management, Anchorage, AK 99508, USA
5. U.S. Fish and Wildlife Service, Portland, OR 97266, USA
6. Monterey Bay Aquarium, Monterey, CA 93940, USA
7. U.S. Geological Survey, Anchorage, AK 99508, USA
8. National Park Service, Fairbanks, AK 99709, USA
* Correspondence: lbowen@usgs.gov

**Abstract:** An opportunistic investigation into ecosystem instability in Kachemak Bay (KBay), Alaska, has led us to investigate exposure to toxic algae in sea otters. We used gene expression to explore the physiological health of sea otters sampled in KBay in May 2019. We found altered levels of gene transcripts in comparison with reference sea otters from clinically normal, oil-exposed, and nutritionally challenged populations sampled over the past decade. KBay sea otters were markedly divergent from the other groups for five genes, which indicated the involvement of neurological, cardiac, immune, and detoxification systems. Further, analyses of urine and fecal samples detected domoic acid in the KBay sea otters. In combination, these results may point to chronic, low-level exposure to an algal toxin, such as domoic acid. With a warming climate, the frequency and severity of harmful algal blooms in marine environments is anticipated to increase, and novel molecular technologies to detect sublethal or chronic exposure to algal toxins will help provide an early warning of threats to the stability of populations and ecosystems.

**Keywords:** sea otter; transcriptomics; domoic acid; Kachemak Bay

## 1. Introduction

Kachemak Bay (KBay) is a subarctic estuary in southcentral Alaska that provides habitat for fish, shellfish, marine mammal, and bird species, and supports important recreational, subsistence, and commercial fisheries. As such, KBay has been named a State of Alaska Critical Habitat Area, and is the largest National Estuarine Research Reserve (NERR) nationwide (https://www.habitatblueprint.noaa.gov/habitat-focus-areas/kachemak-bay-alaska/, accessed on 2 March 2021). However, as with other coastal areas, KBay is at risk of changing climate conditions, including the potential for harmful algal blooms (HABs) and other hazards [1]. Linkages between the health of marine ecosystems and the health of humans and animals that depend on these ecosystems are widely recognized, which has led to the "One Ocean-One Health" research model [2–4]. A key component of this approach is the use of marine mammals as sentinels for potential emerging threats [3,4].

Ideal sentinel species have relatively long lifespans, exhibit high site fidelity, and feed at higher trophic levels, thus representing and consolidating ecosystem perturbations at identifiable spatial scales. Historically, marine mammal sentinels have been used

to identify critical threats to humans and ecosystems from chemical contaminants, but that focus is expanding to include other threats, including those associated with climate change. Harmful agal blooms (HABs) and emerging or reemerging diseases have increased in frequency in conjunction with climate change [5,6] and may be associated with increasing mass mortalities in marine mammals [7]. Marine mammals are an ideal sentinel for the identification of infectious diseases, the effects of contaminants, and the increased frequency and effects of HABs, all of which have direct or indirect human health implications [3,5]. As an alternative to measuring threats directly (for HABs in prey, for contaminants in prey or tissues, and for pathogens), the use of sentinels provides an alternate viewpoint to assess cumulative impacts from multiple threats and, in many cases, to identify physiological and mechanistic processes that can be linked to other species, including humans.

Sea otters (*Enhydra lutris*) are well-established sentinel species of nearshore ecosystems [8–11]. Sea otters were extirpated from KBay by the end of the fur harvest (around 1900) but began to reoccupy the bay in the mid-1980s. By 2012, they had reoccupied KBay (estimated ~6000 sea otters), and numbers have remained high [12]. The increasing sea otter population and resulting effects of density dependence likely contributed to a proliferation of sea otter mortalities, which in 2006 triggered the declaration of the Unusual Mortality Event (UME) [13]. Causes of this UME have been identified as streptococcal endocarditis, encephalitis and/or septicemia, but the mechanisms underlying sea otter susceptibility to these diseases have not been discerned [14]. Speculation of the causes of susceptibility to *Strep* syndrome has included coinfection (*Morbillivirus* and *Bartonella*), overlap with the Pacific Marine Heatwave (PMH), and exposure to harmful algal blooms, all of which can compromise immune system function in aquatic organisms, including sea otters [14–16]. There is emerging evidence that climate change may already be impacting the frequency and severity of harmful algal blooms (HABs) in marine environments [17–20]. Prior to the PMH in 2014–2016 [21], marine biotoxins were detected in 29% of sea otter carcasses (including 26% with domoic acid and 20% with saxitoxin) [14]. Warming waters are likely to expand the geographic range and duration of conditions favorable to algal blooms [5]; thus, algal toxins are a growing concern for sea otters as well as other species in Alaskan marine food webs.

Domoic acid is an algal neurotoxin produced by *Pseudo-nitzschia*, a widely distributed algal species. The standard approach for evaluating the risk of domoic acid is to quantify the abundance of *Pseudo-nitzschia* and domoic acid levels in shellfish [22]; this testing is routinely carried out in many coastal areas where shellfish are harvested for human consumption. In marine mammals, documenting acute domoic acid exposure has included biochemical testing, histopathology, and the observation of neuroexcitation clinical signs. However, these methods are insufficient for assessing chronic sublethal exposures, and the actual extent of domoic acid exposure in marine mammals is likely underrepresented [23,24]. Identification of chronic, low-level exposure to domoic acid is critical to our understanding of nearshore ecosystem health for wildlife and human populations. Thus, an alternative approach for assessing domoic acid exposure is needed to measure the influence of domoic acid on the welfare of the individual, assess physiological reactions, and extend the scope of interpretation to the ecosystem.

Gene-based health diagnostics provide an opportunity for an alternate, whole system, or holistic assessment of health not only in individuals or populations but potentially in ecosystems [25]. Although rapid advances in "-omic" technologies offer unparalleled opportunities to transform the study of HABs, few studies have used this potential [5]. Fewer studies have used "-omics" techniques to study the effects of HABs on health [23,26–28]. Increasingly, gene expression–based diagnostics are being used to monitor ecosystem and wildlife population health [29,30]. Gene expression is physiologically driven by intrinsic and extrinsic stimuli, including toxins, infectious agents, contaminants, trauma, or nutrition. As key indicators of pathophysiologic status, the earliest observable signs of health impairment are altered levels of gene transcripts, evident prior

to clinical manifestation [31], thus providing an early warning of the potentially compromised health of individuals, populations, and ecosystems [32]. Gene-based techniques have the ability to improve our understanding of the subtle effects of chronic environmental perturbations by measuring wildlife's physiological responses.

We have developed gene expression as a tool to enhance our understanding of how environmental conditions and stressors may be linked to the health of sea otters [33–35]. Our objective was to evaluate the health of KBay sea otters in comparison to other sea otter populations across the northern Gulf of Alaska. To do this, we identified gene transcript patterns of 13 genes in the KBay sea otters and compared them with transcript profiles of sea otters from previous studies that are thought to be "normal" and those that have been associated with specific environmental perturbations.

## 2. Materials and Methods

### 2.1. Reference Sea Otters

We identified three free-ranging groups of sea otters from our previous studies to use as reference populations for comparison with the KBay 2019 sea otters. These included a clinically normal sea otter population from the Alaska Peninsula (2009; N = 25; AP), a population from western Prince William Sound (WPWS) with potential exposure to lingering oil from the 1989 *Exxon Valdez* oil spill (2006–2008; N = 82; WPWS1), and a population from western Prince William Sound experiencing some level of nutritional deficit due to higher population densities (2010–2012; N = 85; WPWS2) [32,34–36]. Methods for otter capture, sample processing, and analysis were identical among all populations and are detailed below in sections 2.2–2.5.

### 2.2. Kachemak Bay Otters

Twenty adult female sea otters were sampled in KBay, Alaska, between 25 and 30 May 2019. Sea otters were captured and anesthetized, and blood was drawn by jugular venipuncture within 1–2 h of the initial capture. The capture and sedation methods are detailed in Murray et al. [37]. These animals were identified as clinically normal by wildlife veterinarians at the time of blood collection.

### 2.3. Blood Collection and RNA Extraction

A 2.5 mL sample from each sea otter was drawn directly into a PAXgene blood RNA collection tube (PreAnalytiX, Zurich, Switzerland) from either the jugular or popliteal veins and then frozen at −20 °C until extraction of RNA [34]. Rapid RNA degradation and induced expression of certain genes after blood draws have led to the development of methodologies for preserving the RNA expression profile immediately after blood is drawn. The PAXgene tube contains a blend of RNA-stabilizing reagents that protect RNA molecules from degradation by RNases and prevent further induction of gene expression. The RNA from blood in PAXgene tubes was isolated according to the manufacturer's standard protocols, which included an on-column DNase treatment to remove contaminating gDNA (silica-based microspin technology), and the extracted RNA was stored at −80 °C until analysis. We measured the concentration and clarity on a Qubit 3.0 Fluorometer using RNA, DNA, and RNA IQ Assay Kits (Life Technologies, Carlsbad, CA, USA).

### 2.4. cDNA Synthesis

Standard cDNA synthesis was performed on 2 ug of RNA template from each animal. Reaction conditions included 4 units of reverse transcriptase (Omniscript®, Qiagen, Valencia, CA, USA), 1 uM of random hexamers, 0.5 mM of each dNTP, and 10 units of RNase inhibitor in RT buffer (Qiagen, Valencia, CA, USA). Reactions were incubated for 60 min at 37 °C, followed by an enzyme inactivation step of 5 min at 93 °C, and then stored at −20 °C until further analysis.

*2.5. Real-Time PCR*

The genes chosen for expression profile analysis represent multiple physiological systems that play a role in immuno-modulation, inflammation, cell protection, tumor suppression, cellular stress response, xenobiotic metabolizing enzymes, and antioxidant enzymes. These genes can be modified by biological, physical, or anthropogenic impacts and consequently provide information on the general type of stressors present in a given environment (Table 1).

**Table 1.** Genes used in sea otter analyses (from Bowen et al. [32]).

| Gene | Gene Function |
|---|---|
| S9 | Ribosomal Reference Gene [32] |
| HDC | The HDCMB21P gene codes for a translationally controlled tumor protein (TCTP) implicated in cell growth, cell cycle progression, malignant transformation, tumor progression, and in the protection of cells against various stress conditions and apoptosis [38–40]. The upregulation of HDC is indicative of the development or existence of cancer. Environmental triggers may be responsible for the population-based upregulation of HDC. HDC expression is known to increase with exposure to carcinogenic compounds, such as polycyclic aromatic hydrocarbons [41–43]. |
| COX2 | Cyclooxygenase-2 catalyzes the production of prostaglandins that are responsible for promoting inflammation [44]. Cox2 is responsible for the conversion of arachidonic acid to prostaglandin H2, a lipoprotein critical to the promotion of inflammation [45]. The upregulation of Cox2 is indicative of cellular or tissue damage and an associated inflammatory response. |
| CYT | The complement cytolysis inhibitor protects against cell death [46]. The upregulation of CYT is indicative of cell or tissue death. It is now believed that domoic acid-induced altered $Ca^{+2}$ homeostasis is key in excitotoxic apoptosis, which is consistent with our finding of significantly higher levels of CYT in KBay otters [47]; increased levels of CYT have also been associated with cardiomyopathy [48,49]. |
| AHR | The arylhydrocarbon receptor responds to classes of environmental toxicants including polycyclic aromatic hydrocarbons, polyhalogenated hydrocarbons, dibenzofurans, and dioxin [50]. Depending upon the ligand, AHR signaling can modulate T-regulatory ($T_{REG}$) (immune-suppressive) or T-helper type 17 ($T_H17$) (pro-inflammatory) immunologic activity [51,52]. Wang et al. [53] were the first to identify substantial activation of AHR by domoic acid exposure in fish, a transcriptional response of phase I XME through ligand-activated AHR and ARNT to domoic acid exposure. AHR binds to toxins, initiating a detoxification cascade and an altered immune response. Activation of the AHR pathway also contributes to cardiac malformation [54]. |
| THRB | The thyroid hormone receptor beta can be used as a mechanistically based means of characterizing the thyroid-toxic potential of complex contaminant mixtures [55]. Thus, increases in THR expression may indicate exposure to organic compounds, including PCBs, and associated potential health effects, such as developmental abnormalities and neurotoxicity [55]. Hormone-activated transcription factors bind DNA in the absence of hormones, usually leading to transcriptional repression [56]. |
| HSP70 | The heat shock protein 70 is produced in response to thermal or other stress [57,58]. In addition to being expressed in response to a wide array of stressors (including hyperthermia, oxygen radicals, heavy metals, and ethanol), heat shock proteins act as molecular chaperones [59]. For example, heat shock proteins aid in the transport of the AHR/toxin complex in the initiation of detoxification [60]. |
| IL-18 | Interleukin-18 is a pro-inflammatory cytokine [44]. It plays an important role in inflammation and host defense against microbes [61]. |
| IL-10 | Interleukin-10 is an anti-inflammatory cytokine [44]. Levels of IL-10 have been correlated with the relative health of free-ranging harbor porpoises, e.g., increased amounts of IL-10 correlated with chronic disease, whereas the cytokine was relatively reduced in apparently fit animals experiencing acute disease [62]. The association of IL-10 expression with chronic disease has also been documented in humans [63]. |
| DRB | A component of the major histocompatibility complex, the DRB class II gene, is responsible for the binding and presentation of processed antigen to $T_H$ lymphocytes, thereby facilitating the initiation of an immune response [44,64]. The upregulation of MHC genes has been positively correlated with parasite load [65], whereas the downregulation of MHC has been associated with contaminant exposure [66]. |
| Mx1 | The Mx1 gene responds to viral infection [67]. Vertebrates have an early strong innate immune response against viral infection, characterized by the induction and secretion of cytokines that mediate an antiviral state, leading to the upregulation of the MX-1 gene [68]. |
| CCR3 | The chemokine receptor 3 binds at least seven different chemokines, and it is expressed on eosinophils, mast cells |

| | | |
|---|---|---|
| | (MC), and a subset of Th cells (Th2) that generate cytokines implicated in mucosal immune responses [69,70]. The upregulation of CCR3 occurs in the presence of parasites [69,70]. | |
| HTT5 | The serotonin transport gene codes for an integral membrane protein that transports the neurotransmitter serotonin from the synaptic spaces into presynaptic neurons. This transport of serotonin by the SERT protein terminates the action of serotonin and recycles it in a sodium-dependent manner [71,72]. Algal toxins have been associated with increased expression of HTT5 [73]; at the cellular level, domoic acid is an excitatory amino acid analogue of glutamate, a major excitatory neurotransmitter in the brain known to activate glutamate receptors [74]. | |
| CaM | Calmodulin (CaM) is a small acidic Ca2+-binding protein, with a structure and function that is highly conserved in all eukaryotes. CaM activates various Ca2+-dependent enzyme reactions, thereby modulating a wide range of cellular events, including metabolism control, muscle contraction, exocytosis of hormones and neurotransmitters, and cell division and differentiation [75]. CaM has also been reported to be a pivotal calcium metabolism regulator in shell formation [76]. Algal toxicity is associated with increased intracellular $Ca^{+2}$ [74,77–82]. This intracellular excess is toxic to the cells and triggers the activation of several detrimental cascading effects [47]. | |

Real-time PCR systems for the individual, sea otter-specific reference or housekeeping gene (S9) and genes of interest were run in separate wells ((Table 2) (Bowen et al. [34], Miles et al. [35]). Briefly, 1 uL of cDNA was added to a mix containing 12.5 uL of Quanti-Tect SYBR Green Master Mix [5 mM Mg2+] (Qiagen, Valencia, CA, USA), 0.5 uL each of forward and reverse sequence specific primers, and 10.0 uL of RNase-free water; the total reaction mixture was 25 uL. The reaction mixture cDNA samples for each gene of interest and the S9 gene were loaded into 96-well plates in duplicate and sealed with optical sealing tape (Applied Biosystems, Foster City, CA, USA). Reaction mixtures containing water but no cDNA were used as negative controls; thus, approximately 3–4 individual sea otter samples were run per plate.

**Table 2.** Sea otter-specific quantitative real-time polymerase chain reaction primers were used in the analysis of free-ranging otters.

| Gene | Primer Name | FP1 | Primer Name | RP1rc | Expected Amplicon (bp) |
|---|---|---|---|---|---|
| HDC | Enlu HDC F1 | ATGTTCTCCGACATCTACAAGATCC | Enlu HDC R1rc | GTTTCCTGCAGGTGATGGTTCATG | 82.6 |
| COX2 | Mv COX2 F1 | CATTCCTGATCCCCAGGGCAC | Mv COX2 R1rc | GTCCACCCCATGGCCCAGTC | 79.4 |
| CYT | Mv CYT F1 | GCTGGACGAGCAGTTTAGCTGG | Mv CYT R1rc | GACGCCAGAGGGAGCACTGG | 81.8 |
| AHR | Enlu AHR F1 | GCGCTGAGTACCATATACGGATGA | Enlu AHR R1rc | CACTAAGCGTGCATTAGACTGAAC | 76.8 |
| THRB | Mv THRB F1 | GGACAAACCGAAGCACTGTCCAG | Mv THRB R1rc | GGAATATYGAGCTAAGTCCAAGTGG | 81.8 |
| HSP70 | Mv HSP70 F | CCAGGTGGCGCTGAACCCGC | Mv HSP70 Rrc | CCTTGTAGCTCACCTGCACCTTG | 85.6 |
| IL-18 | Mv IL-18 F | GTACAGAAAACGCATCCCATACC | Mv IL-18 Rrc | CTGGAGGTCTCATTTCCTTAAAGG | 76.2 |
| IL-10 | Mv IL-10 F2 | GACTTTAAGRGTTACCTGGGTTGC | Mv IL-10 R2rc | TCCACSGCCTTGCTCTTRTTYTC | 83.7 |
| DRB | Mv DRB F | CGGCGAGTGGAGCCTATAGTG | Mv DRB Rrc | CTCCTCTTCCTGGCCATTCCG | 81.0 |
| Mx1 | Enlu Mx1 F | CAAGCAGCTCATCAGGAAGTACA | Enlu Mx1 Rrc | GGTGGCGATGTCCACGTT | 79.5 |
| S9 | Mv S9 F | CCAGCGCCACATCAGGGTCCG | Mv S9 Rrc | CCCTGGCCTTTCTTGGCGTTC | 83.4 |

| CCR3 | Enlu CCR3 F | CTGCTGGGCAATGTGGTGG TG | Enlu CCR3 Rrc | GAAAGAGCAAGTCAGAAAT GGCC | 106 |
|------|------------|------------------------|---------------|--------------------------|-----|
| CaM | Enlu CaM F1 | GCGAGGCATTCCGAGTC | Enlu CaM R1rc | TCTGATCATTTCATCTACTTC T | 122 |
| HTT5 | Enlu HTT5 F2 | TCCTCCTGCCCTACACGA | Enlu HTT5 R2rc | CGGTGGTACTGGCCCAG | 82 |

Amplifications were conducted on a Step-One Plus Real-time Thermal Cycler (Applied Biosystems, Foster City, CA, USA). Reaction conditions were as follows: 50 °C for 2 min, 95 °C for 15 min, 40 cycles of 94 °C for 30 s, 60 °C for 30 s, 72 °C for 31 s, an extended elongation phase at 72 °C for 10 min. Reaction specificity was monitored by melting curve analysis using a final data acquisition phase of 60 cycles of 65 °C for 30 s and verified by direct sequencing of randomly selected amplicons [41]. The cycle threshold crossing values (CT) for the genes of interest were normalized to the S9 housekeeping gene.

### 2.6. Analysis of Urine and Feces for the Presence of Domoic Acid

Urine was collected by cystocentesis, and feces were collected either by free catching, manually from the rectum, or opportunistically during processing. Samples were immediately frozen and stored at −20 °C until toxin analyses were performed. Domoic acid was extracted from urine and feces, as described by Lefebvre et al. [83]. Domoic acid was quantified in filtered extracts using a commercially available domoic acid competitive ELISA kit, following the instruction protocol supplied by the manufacturer (Biosense® Laboratories, Bergen, Norway, and Abraxis LLC, Warminster, PA, USA), with minor modifications described in Lefebvre et al. [83]. The limits of detection for domoic acid in the sample material were 4 ng/g or ml for feces and 0.4 ng/mL urine.

### 2.7. Statistical Analysis

Analyses of qPCR data used normalized $C_T$ (threshold crossing) values (housekeeping gene $C_T$ subtracted from the gene of interest $C_T$); the lower the normalized value, the more transcripts were present. A change in the normalized value of 2 is approximately equivalent to a 4-fold change in the amount of the transcript. We calculated the means and standard deviations for all genes within each population. Differences in means were calculated using a Kruskal–Wallis non-parametric rank ANOVA test, which included Dunn's Test for multiple comparisons (NCSS, Statistical and Power Analysis Software, Kaysville, UT, USA). We then analyzed separation among areas (AP, KBay, WPWS1, WPWS2) and identified genes influencing separation using multivariate discriminant function analysis (DFA) (NCSS, Statistical and Power Analysis Software, Kaysville, UT, USA). We used Pearson correlations to test for the strength of relationships between fecal domoic acid levels (ng/g) and gene expression levels (normalized $C_T$) (NCSS, Statistical and Power Analysis Software, Kaysville, UT, USA).

## 3. Results

### 3.1. Gene Expression Analyses

Real-time PCR data are represented as normalized values (NVs); the lower the NV, the higher the quantity of transcripts. The means and standard deviations for all genes within each population were calculated (Table 3). We identified the distribution of the average normalized threshold crossing ($C_T$) values across genes targeted by a panel of 13 primer pairs (Figure 1). We identified differences in means among areas for all but two genes (IL-18 and CCR3) using the Kruskal–Wallis non-parametric rank ANOVA test (NCSS, Statistical and Power Analysis Software, Kaysville, UT, USA). Genes for which KBay otters are most divergent from the other groups are CYT, AHR, THRB, HTT5, and CaM.

**Table 3.** Means, standard deviations (CT values), and ANOVA *p* values for sea otters sampled in Kachemak Bay (KBay), the Alaska Peninsula, and Western Prince William Sound (WPWS1 and WPWS2). Letter (a,b,c,d) differences denote significant differences among populations (Kruskal–Wallis with Dunns' multiple comparison); lack of a letter (a,b,c,d) denotes no significant difference from any other group. Note: lower numbers indicate higher expression levels. Gene abbreviations and descriptions are provided in Table 1.

| Genes | Kachemak Bay (2019) | Alaska Peninsula (2009) | WPWS1 (2006–2008) | WPWS2 (2010–2012) | ANOVA *p* value |
|---|---|---|---|---|---|
| | N = 20 | N = 27 | N = 81 | N = 87 | |
| | Mean (SD) | Mean (SD) | Mean (SD) | Mean (SD) | |
| HDC | 10.68 [a] (1.21) | 6.39 [b] (1.73) | 1.05 [b] (4.79) | 9.18 [a] (1.85) | 0.00 |
| COX2 | 7.73 [a,b] (2.00) | 6.77 [b] (1.78) | 8.19 [a] (1.76) | 9.44 [c] (1.53) | 0.00 |
| CYT | −2.64 [a] (12.10) | 2.29 [b] (1.28) | 1.61 [b] (2.30) | 1.76 [b] (1.02) | 0.00 |
| AHR | −2.41 [a] (14.21) | 10.64 [a] (1.89) | 10.41 [a] (2.38) | 12.16 [b] (1.39) | 0.00 |
| THRB | 16.46 [a,c] 4.57 | 13.40 [a] 3.20 | 11.87 [b] 3.32 | 16.27 [c] 2.81 | 0.00 |
| HSP70 | 12.42 [a,c] (1.07) | 8.65 [b] (1.57) | 10.11 [b] (2.30) | 13.91 [c] (2.56) | 0.00 |
| IL-18 | 3.55 [a] (1.22) | 2.65 [b,c] (4.42) | 1.91 [c] (3.02) | 2.45 [b] (1.02) | 0.063 |
| IL-10 | 19.26 [a,c] (5.10) | 13.32 [b] (2.96) | 13.75 [b] (2.96) | 20.69 [c] (3.78) | 0.00 |
| DRB | 2.42 [a] (5.12) | −0.87 [c] (1.54) | 0.37 [b,d] (1.46) | −0.073 [c,d] (0.83) | 0.00 |
| MX1 | 13.47 [a,c] (1.18) | 12.90 [a] (3.62) | 10.58 [b] (1.65) | 15.17 [c] (2.35) | 0.00 |
| CCR3 | 4.31 (1.17) | 5.31 (2.13) | 5.16 (1.40) | 5.11 (1.15) | 0.074 |
| HTT5 | −1.03 [a] (0.92) | 9.76 [b] (1.49) | 9.99 [b] (1.27) | 11.07 [c] (2.20) | 0.00 |
| CaM | 7.48 [a] (3.34) | −0.093 [b] (0.70) | −1.75 [c] (0.92) | −0.67 [b] (0.40) | 0.00 |

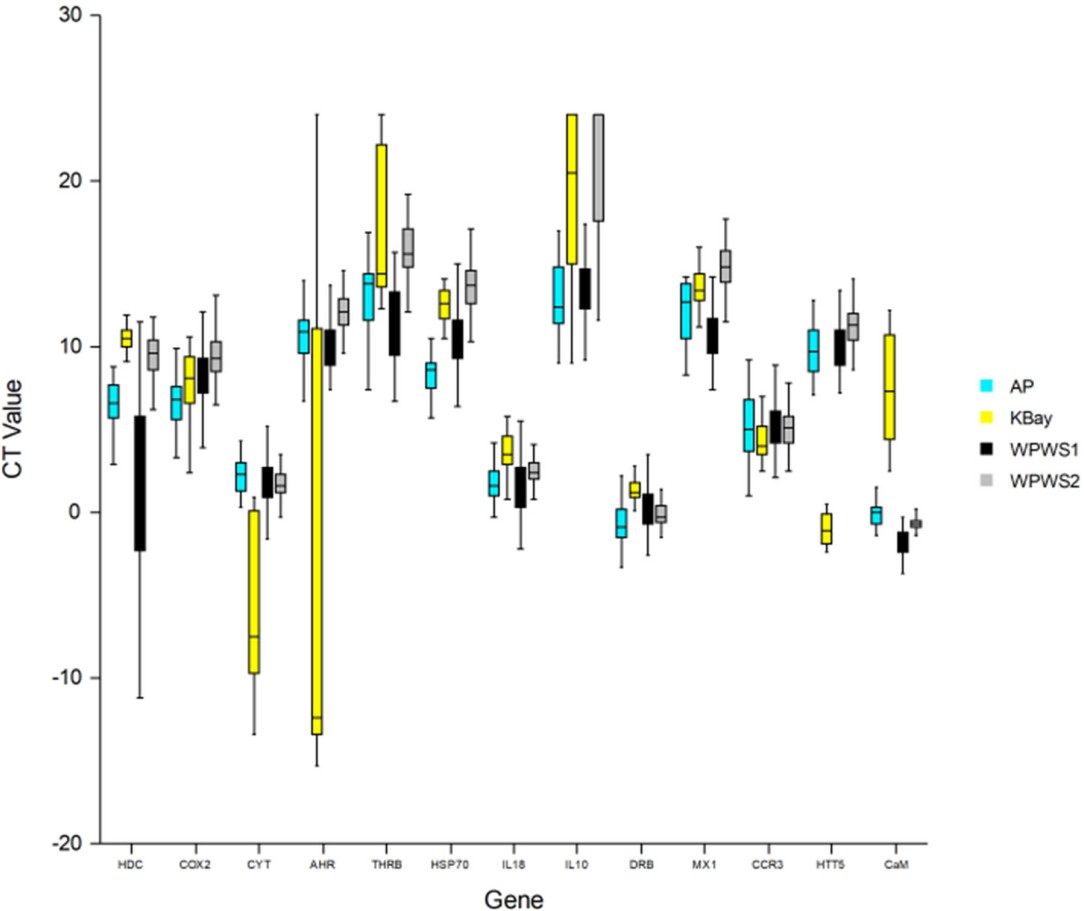

**Figure 1.** Real-time PCR data on 13 genes sampled in sea otters. Normalized values (NVs) are presented; the lower the NV, the higher the quantity of transcripts (i.e., smaller values = more transcripts). Boxes are delineated by the 25th and 75th percentiles. The 50th percentile median is indicated. Whisker length uses the classic method of box edge + (1.5; interquartile range), and outliers have been removed. Gene abbreviations are provided in Table 1. Blood was sampled in sea otters in Alaska Peninsula (AP) (2009), Kachemak Bay (KBay) (2019), Western Prince William Sound (WPWS1) (2006–2008), and WPWS2 (2010–2012). We identified differences in means for all but two genes (IL-18 and CCR3) using the Kruskal–Wallis non-parametric rank ANOVA test (NCSS, Statistical and Power Analysis Software, Kaysville, UT, USA). Genes for which KBay otters are most divergent from the other groups are CYT, AHR, THRB, HTT5, and CaM.

Discriminant function analysis of transcript patterns identified a clear separation of KBay sea otters from the other groups (AP, WPWS1, WPWS2) (Figure 2). Consistent with the ANOVA results, all genes except HSP70 and CCR3 contributed to the separation of populations.

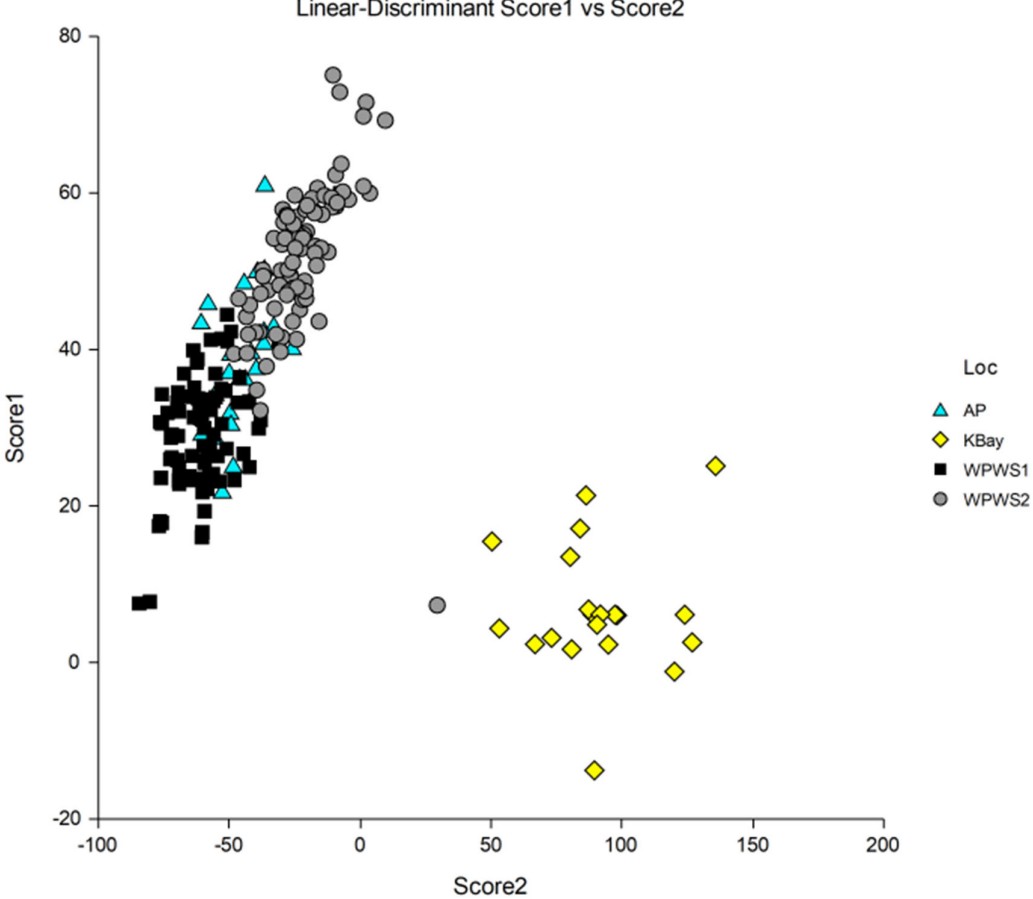

**Figure 2.** Discriminant function analysis of transcript patterns of sea otters from 4 areas, including Kbay. Each data point corresponds to an individual sea otter and represents the centroid of gene expression levels as a function of discriminant variables 1 and 2, which maximize individual separation.

### 3.2. Analysis of Urine and Feces for the Presence of Domoic Acid

Urine samples from four otters and fecal samples from eight otters were analyzed for the presence of domoic acid. Domoic acid was detected in all four urine samples, and seven of the eight fecal samples were analyzed via enzyme-linked immunosorbent assays (ELISA). Domoic acid concentrations ranged from 0.7 to 20 ng/mL in urine and from 38 to 651 ng/g in feces (Supplementary Spreadsheet S1). No significant correlations were found between gene expression levels and levels of domoic acid in feces (Table 4).

**Table 4.** Pearson correlations between levels of fecal domoic acid (ng/g) and gene expression (normalized CT).

| Gene | Pearson Correlation | *p* Value |
|---|---|---|
| HDC | 0.39 | 0.39 |
| COX2 | 0.097 | 0.84 |
| CYT | −0.31 | 0.50 |
| AHR | 0.73 | 0.062 |
| THRB | 0.18 | 0.69 |
| HSP70 | 0.52 | 0.23 |

| | | |
|---|---|---|
| IL-18 | 0.31 | 0.49 |
| IL-10 | 0.035 | 0.94 |
| DRB | 0.53 | 0.22 |
| Mx1 | 0.16 | 0.73 |
| CCR3 | 0.74 | 0.057 |
| CaM | 0.23 | 0.62 |
| HTT5 | 0.12 | 0.80 |

## 4. Discussion

This is an initial examination of gene transcript data from sea otters captured in May 2019 in KBay, Alaska. We have chosen to compare transcript levels of KBay otters with those from sea otters previously captured in nearby areas (AP in 2009, WPWS1 in 2006–2008, and WPWS2 in 2010–2012), as these groups represent a range of transcriptomic responses. The Alaska Peninsula otters represent "healthy" sea otters that were not subject to substantial hydrocarbon exposure, disease, or food limitations. These interpretations are further supported by population status and trajectory data, indicating that the AP population of sea otters was below the carrying capacity when sampled in 2009 [32,33,84]. Transcript profiles in WPWS2 otters are characterized by relatively low levels of expression across the gene panel. Low levels of T-cell-linked gene transcripts may indicate a nutritional deficit [85] or an unbalanced physiological resource allocation. This interpretation is consistent with data on energy intake rates of various sea otter populations, indicating that food resources for sea otters in WPWS2 were relatively limited [84], and it is also supported by the population status of stable or near carrying capacity for WPWS2 [32]. The WPWS1 sea otters were sampled at a time when exposure to lingering oil from the *Exxon Valdez* oil spill was still a concern [86]. These sea otters appeared to have immunological or physiological responses that indicated greater organic compound exposure relative to the other groups examined (Alaska Peninsula, Katmai, Kodiak, WPWS2), consistent with a polycyclic aromatic hydrocarbon (PAH)-induced profile [32]. Thus, these three groups provide a useful context for evaluating the KBay gene expression results.

Domoic acid is a marine neurotoxin produced by some diatom species of the genus *Pseudo-nitzschia.* Although large-scale mortality events have been reported in marine mammal and sea bird populations due to the consumption of domoic acid-contaminated fish [22,87–91], there have been no historical reports of major toxic events in Alaska related to domoic acid. In California, however, marine mammal mortalities associated with domoic acid toxicity occur yearly [20], and more effort has been directed toward understanding the mechanisms and effects of domoic acid toxicity. Acute, subacute, and chronic domoic acid exposure can elicit a wide range of physiological impacts [24,74,92,93].

Acute domoic acid poisoning is characterized by a constellation of clinical symptoms and signs involving multiple organ systems, including the gastrointestinal tract, the central nervous system (CNS), and the cardiovascular system [74]. Domoic acid has a high affinity for glutamate receptors [94]. After binding to glutamate receptors, domoic acid induces excitotoxicity through cell depolarization and a subsequent increase in intracellular calcium and apoptosis [95]. A widely accepted route of pathogenesis is the initial direct interaction with the heart [96], as domoic acid engages glutamate receptors in the mammalian heart, causes calcium-associated perturbation, and results in apoptotic damage [96]. Chronic asymptomatic domoic acid poisoning appears to cause the same physiological perturbations as acute domoic acid poisoning but at subclinical levels, making observational detection more difficult [93].

We identified five genes in KBay otters with markedly divergent expression levels from otters in the other three areas: AHR, HTT5, THRB, CaM, and CYT. In comparison with the other groups, the KBay otters exhibited extremely high levels of AHR expres-

sion. The AHR pathway is a well-characterized response mechanism of crude oil exposure that contributes to cardiac malformations [54]. At the most basic level, exposure of an organism to a hydrocarbon leads to immediate changes in gene expression related to xenobiotic metabolism, as well as changes in expression that may lead to impairment of important biological processes and abnormal development [47]. Mammalian cytochrome P450 1A (CYP1A) expression is regulated by the ligand-activated transcription factor, AHR, and its dimerization partner, the aryl hydrocarbon receptor nuclear translocator (ARNT) [53]. The KBay sea otters had transcript profiles very different from those of otters with suspected oil exposure (WPWS1), and in fact, oil exposure is not suspected to be a factor influencing transcript profiles in KBay, although we did not measure levels of oil or oil components in KBay sea otters. An alternative hypothesis is that increased levels of AHR may be linked to exposure to harmful algal bloom toxins. Although Washburn et al. [97] documented the induction of the AHR/ARNT complex in fish exposed to brevetoxin, Wang et al. [53] were the first to identify substantial activation of AHR, ARNT, and CYP1A by domoic acid exposure in fish, a transcriptional response of phase I XME through ligand-activated AHR and ARNT to domoic acid exposure. AHR binds to toxins, initiating a detoxification cascade and an altered immune response.

In addition to AHR, we identified very high levels of HTT5 (sodium-dependent serotonin transporter) in KBay sea otters. HTT5 is a serotonin transport gene coding for an integral membrane protein (SERT) that transports the neurotransmitter serotonin from the synaptic spaces into presynaptic neurons. Transport of serotonin by the SERT protein terminates the action of serotonin and recycles it in a sodium-dependent manner [71,72]. Coincidentally, algal toxins have been shown to be excitatory to neurotransmitters and receptors [28,27]. Pazos et al. [73] identified increased expression of HTT5 in mussels (*Mytilus galloprovincialis*) exposed to domoic acid. At the cellular level, domoic acid is an excitatory amino acid analogue of glutamate, a major excitatory neurotransmitter in the brain, and is known to activate glutamate receptors [74]. The molecular targets for glutaminergic neurotransmission are ubiquitously expressed in the heart and other tissues [74,98–101], providing evidence that the molecular targets for excitatory neurotransmission and neurotoxicity of excitatory amino acids are present in the human heart. In fact, the heart was one of the most affected organs in sea lions that died of domoic acid intoxication [102,74]. Domoic acid exposure has been identified as a risk factor associated with myocarditis and dilated cardiomyopathy in southern sea otters [24,74,103].

Another effect of domoic acid toxicity is increased intracellular $Ca^{2+}$ [74,77–82]. This intracellular excess is toxic to the cells and triggers the activation of several detrimental cascading effects [47]. Altered cardiac gene expression, particularly for genes involved in intracellular $Ca^{+2}$ homeostasis and excitation–contraction coupling, may directly contribute to cardiotoxicity [104,47]. It is now believed that domoic acid-induced altered $Ca^{+2}$ homeostasis is key to excitotoxic apoptosis, which is consistent with our finding of significantly higher levels of CYT in KBay sea otters [47]. Increased levels of CYT have also been associated with cardiomyopathy [48,49]. Additionally, CaM levels in KBay sea otters were significantly lower than in reference otters, which is consistent with high intracellular $Ca^{+2}$ levels. Studies of California sea otters have shown an association between cardiomyopathy and domoic acid exposure [24]. We also identified relatively low levels of thyroid hormone receptor beta, THRB, in KBay sea otters. Serum thyroid hormones are altered by several excitatory amino acids, including domoic acid, suggesting that these compounds can modulate the regulation of hormone secretion from the pituitary–thyroid axis [74,105,106], which may explain the low THRB transcript levels in KBay sea otters.

In combination, these marked differences in gene transcript levels in KBay sea otters relative to sea otters in other areas of Alaska may point to chronic, low-level exposure to an algal toxin, such as domoic acid. Similar to what was seen in the 2014–2016 PMH, water temperatures in KBay were unusually warm during the summer of 2019

[107]. Water quality sampling in KBay in 2019 confirmed the presence of the toxin-producing algae genera *Dinophysis*, *Alexandrium*, and *Pseudo-nitzschia* in most months of the year, with *Pseudo-nitzschia* dominant from July to August. Although no HABs were noted in the main part of KBay, blooms were observed in several sub-bays [108]. It is important to note that because sea otters consume large amounts of benthic filter-feeding invertebrates that bioconcentrate domoic acid, they are indicators of domoic acid trophic transfer [24]. It seems plausible, especially in consideration of domoic acid detection in urine and fecal samples, that sea otters in KBay were experiencing chronic exposure to low levels of domoic acid toxins.

It is understood that climate change will impact the frequency and severity of HABs in marine environments [17–19], as warming waters will expand the range and duration of conditions favorable to algal blooms [109,110]. Although the acute effects of high domoic acid exposure are well known [47], little information exists on the effects of repeated, low doses of domoic acid on mammals [24,93,94]. More recently, the lasting health effects of repeated domoic acid exposure in humans and wildlife have been recognized [24]. Identification of organisms experiencing chronic, low-level exposure to domoic acid is critical to our understanding of nearshore ecosystem health for wildlife and human populations. Our data indicate that, in fact, sea otters in KBay are exposed to sublethal doses of domoic acid, as 100% of the urine samples ($n$ = 4 individuals) and 88% of the fecal samples ($n$ = 8 individuals) analyzed from live sea otters in 2019 contain the toxin. Further, Burek Huntington et al. [14] also found~25% of sea otter carcasses tested for HABs had detectable levels of domoic acid and saxitoxin. However, there is no way to determine the actual domoic acid doses that these animals are experiencing based on fecal and urine domoic acid concentrations, as these samples are only a snapshot in time, with no information on the duration of exposure and uptake and depuration rates. The domoic acid concentrations quantified here are in the range of those measured in urine and fecal samples from California sea lions (*Zalophus californianus*) with known domoic acid toxicosis (Figure 2 in Lefebvre et al. [83]); however, it is not known what the total dose or duration of exposure was for any of the sea otters in our study. In addition, although in a functional system we expect a transcriptional response from certain genes upon exposure to a corresponding stressor due to the complexities of the integrated signaling networks involved, we would not necessarily anticipate a linear relationship between fecal domoic acid level and the quantity of gene transcript. With exposure, there will likely be an immediate (minutes) and synchronous activation of multiple genes, followed by restriction of adaptive responses (negative feedback loops to avoid immunopathology). Simultaneously, depuration of the toxin occurs, with the end result of a very complex relationship between gene transcripts and the presence of a toxin, such as domoic acid. Domoic acid concentrations quantified at a single time point in naturally exposed animals do not depict total exposure or duration of exposure.

In the future, the advent of single-cell analysis should "provide us with a comprehensive view of basic characteristics such as the presence of thresholds and signal noise in gene expression" [111], but at present, in follow-up studies, the goal will be to refine a more general exposure/response model. The fact that the live sea otters appeared to be healthy and did not experience overt excitotoxic behaviors suggests that exposure is low and subacute. Further studies examining domoic acid dynamics in sea otter prey species would help identify sublethal mechanisms and/or avenues of toxicity.

The gene expression differences observed between the KBay sea otters and those from other areas suggest significant physiological perturbations in the KBay population. The expression levels of these five genes can be modified by more than one environmental stressor (Table 1). This, in combination with the lack of extensive environmental data, precludes us from making definitive linkages between domoic acid and gene expression patterns. However, the patterns of altered gene transcripts and physiological systems affected, in conjunction with the detection of domoic acid in urine and fecal samples, point toward exposure to an algal toxin. Subsequently, the role of domoic acid as a cause of harmful physiological changes, and its routes, duration, and severity of exposure warrant further investigation.

**Supplementary Materials:** The following supporting information can be downloaded at: www.mdpi.com/xxx/s1, Spreadsheet S1.

**Author Contributions:** Conceptualization: S.K., K.K., and K.L.; Methodology: S.K., K.L., and L.B.; Formal Analysis: L.B. and S.W.; Data Curation: S.W.; Writing—Original Draft Preparation: L.B., B.W., B.B., H.C. S.K., K.L., and M.S.M.; Writing—Review and Editing: M.M., K.K., D.M., B.B., H.C., S.W., and C.C.; Project Administration: C.C. All authors have read and agreed to the published version of the manuscript.

**Funding:** Principal funding for the capture and collection of samples that occurred in Kachemak Bay was provided by the USFWS Marine Mammals Management. USGS Ecosystems Mission Area provided funding for gene expression analysis. Support for algal toxin analyses was provided by ECOHAB project number NA20NOS4780195 (to KAL; ECOHAB contribution #1017).

**Institutional Review Board Statement:** Kachemak Bay field activities were conducted under the Marine Mammal Protection Act USFWS permit (MA041309)

**Informed Consent Statement:** Not applicable.

**Data Availability Statement:** The data presented in this study are available in Supplementary Spreadsheet S1.

**Acknowledgments:** We appreciate contributions made to this study. The findings and conclusions in this article are those of the authors and do not necessarily represent the views of the U.S. Fish and Wildlife Service.. We thank all of the field assistance—specifically, George Esslinger, Joe Tomoleoni, Bill Beatty, Luke Porter, Markus Horning, and Alaska Sea Life Center Veterinary Staff for their sea otter sampling efforts, Alaska Maritime National Wildlife Refuge, NOAA Kasitsna Bay Laboratory, and the R/V Alaskan Gyre for their services and support during the project. Any use of trade, firm, or product names is for descriptive purposes only and does not imply endorsement by the U.S. Government.

**Conflicts of Interest:** The authors declare no conflict of interest.

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
