# Peer review of "Divergent Gene Expression Profiles in Alaskan Sea Otters: An Indicator of Chronic Domoic Acid Exposure?"

_2673-1924, doi:10.3390/oceans3030027_

Round 1
Reviewer 1 Report
A very interesting monitoring article by American researchers rightly points out the insufficiency of using the currently information-intensive approach - transcriptomics - to study global environmental changes in order to assess the risks of exposure to certain harmful factors. The results of this work will undoubtedly be of interest not only to specialists, but also to researchers in general, as they make an additional contribution to the treasury of increasing the importance of using omics technologies in various fields. The manuscript has strengths and weaknesses, and can be recommended for publication after major revision.
Major points
- The introduction is well written, but I feel like there is potential for reduction. Otherwise, it seems excessively large in relation to the rest of the material of the article.
- The volume of introduction could be partially reduced by moving the table with genes of interest and their description to the "Materials and Methods" section. It seems that there it will be more appropriate, for example, as a separate subsection.
- The table of primers used in this study is not presented. Please place it in the “Materials and methods” chapter.
- The results lack a table with data on domoic acid for all individuals. In addition, the fluctuation of levels of domoic acid for both urine and feces is very strong. Based on such a strong scatter, it is difficult to draw serious conclusions on the effect of domoic acid on gene transcription in the population as a whole. It is then necessary to examine each individual for the presence and amount of domoic acid, determine the level of transcription of each gene of interest and look for a correlation between them.
- How would the authors account for the possibility that other sea otter populations might also have some exposure to domoic acid, in addition to the other factors considered here? Also, a potential oil impact on KBay sea otters cannot be ruled out, even if there is no information on possible spills/leaks in this particular area. Organics could be brought into the bay by sea currents. The real impact of domoic acid on the body of a particular sea otter could be assessed, in the opinion of the reviewer, only using the recommendation presented in the previous comment.
Most likely, the team already has all the necessary information, and it would be possible without additional experiments to perform a comparative analysis of the levels of domoic acid in urine and feces and the levels of transcription of certain genes of these particular individuals. Such results could be presented in an additional section, and they would clearly demonstrate the correlation of changes in the expression levels of genes of interest with the presence of domoic acid in the secretions of a particular individual. And although this will also be a very speculative picture, as the authors note in the Discussion, based on a “snapshot”, but it can still supplement the study in some way. After all, taking blood and determining transcripts is also a kind of “snapshot”, since the level of these transcripts can change over time in the presence / absence of a stress factor, and, I believe, very quickly as well.
Author Response
Comments and Suggestions for Authors
A very interesting monitoring article by American researchers rightly points out the insufficiency of using the currently information-intensive approach - transcriptomics - to study global environmental changes in order to assess the risks of exposure to certain harmful factors. The results of this work will undoubtedly be of interest not only to specialists, but also to researchers in general, as they make an additional contribution to the treasury of increasing the importance of using omics technologies in various fields. The manuscript has strengths and weaknesses, and can be recommended for publication after major revision.
Major points
- The introduction is well written, but I feel like there is potential for reduction. Otherwise, it seems excessively large in relation to the rest of the material of the article.
We have reduced the length of the introduction.
- The volume of introduction could be partially reduced by moving the table with genes of interest and their description to the "Materials and Methods" section. It seems that there it will be more appropriate, for example, as a separate subsection.
We have moved Table 1 and general gene descriptions into section 2.5 (Real-time PCR) in the Methods.
- The table of primers used in this study is not presented. Please place it in the “Materials and methods” chapter.
We have added a primer table to the materials and methods section.
- The results lack a table with data on domoic acid for all individuals. In addition, the fluctuation of levels of domoic acid for both urine and feces is very strong. Based on such a strong scatter, it is difficult to draw serious conclusions on the effect of domoic acid on gene transcription in the population as a whole. It is then necessary to examine each individual for the presence and amount of domoic acid, determine the level of transcription of each gene of interest and look for a correlation between them.
The results were included in the supplemental data file. We have inserted a reference to the data file to make that clear: “3.2. Analysis of Urine and Feces for presence of Domoic Acid - Urine samples from four otters and fecal samples from eight otters were analyzed for presence of domoic acid. Domoic acid was detected in all four urine samples and seven of the eight fecal samples analyzed via Enzyme-linked immunosorbent assays (ELISA). Domoic acid concentrations ranged from 0.7 to 20 ng/mL in urine and 38 to 651 ng/g in feces (Supplementary data file).”
Lines 364-370 state: “However, there is no way to determine the actual domoic acid doses that these animals are experiencing based on fecal and urine domoic acid concentrations, as these samples are only a snapshot in time with no information on duration of exposure and uptake and depuration rates. The domoic acid concentrations quantified here are in the range of those measured in urine and fecal samples from California sea lions (Zalophus californianus) with known domoic acid toxicosis (Figure 2 in Lefebvre et al. [84]); however, it is not known what the total dose or duration of exposure was for any of the sea otters in our study.”
To underscore this point, we have also added to the discussion: “In addition, although in a functional system we expect a transcriptional response from certain genes upon exposure to a corresponding stressor, due to the complexities of the integrated signaling networks involved, we would not anticipate a linear relationship between fecal domoic acid level and quantity of gene transcript. With exposure there will be an immediate (minutes) and synchronous activation of multiple genes, followed by restriction of adaptive responses (negative feedback loops to avoid immunopathology). Simultaneously, depuration of the toxin is occurring, with the end result a very complex relationship between gene transcripts and a toxin such as domoic acid. In the future, the advent of single-cell analysis should “provide us with a comprehensive view of basic characteristics such as the presence of thresholds and signal noise in gene expression” (de Nadal et al. 2011), but at present the more general exposure/response model must suffice.”
- How would the authors account for the possibility that other sea otter populations might also have some exposure to domoic acid, in addition to the other factors considered here? Also, a potential oil impact on KBay sea otters cannot be ruled out, even if there is no information on possible spills/leaks in this particular area. Organics could be brought into the bay by sea currents. The real impact of domoic acid on the body of a particular sea otter could be assessed, in the opinion of the reviewer, only using the recommendation presented in the previous comment.
You make an excellent point. In the past, otters we examined were not analyzed for domoic acid levels and thus, we are unable to make that comparison. The extreme divergence in KBay transcript profiles from other otter groups should at least spur further research into the effects of domoic acid on otter gene expression as well as the potential use of gene expression technologies as an early warning indicator of individual, population, and ecosystem health.
We have edited the following: “The KBay sea otters had transcript profiles very different from otters with suspected oil exposure (WPWS1), and in fact oil exposure is not suspected to be a factor influencing transcript profiles in KBay, although we did not measure levels of oil or oil components in KBay sea otters.”
We have also expanded the concluding paragraph to read: “The gene expression differences observed between KBay sea otters and those from other areas suggest significant physiological perturbations in the KBay population. Although the small sample size and limited environmental data preclude definitive linkages, the patterns of altered gene transcripts and physiological systems affected, in conjunction with the detection of domoic acid in urine and fecal samples, point towards exposure to an algal toxin. However, the role of domoic acid as a cause of harmful physiological changes, and its routes, duration, and severity of exposure, warrant further investigation.”
Most likely, the team already has all the necessary information, and it would be possible without additional experiments to perform a comparative analysis of the levels of domoic acid in urine and feces and the levels of transcription of certain genes of these particular individuals. Such results could be presented in an additional section, and they would clearly demonstrate the correlation of changes in the expression levels of genes of interest with the presence of domoic acid in the secretions of a particular individual. And although this will also be a very speculative picture, as the authors note in the Discussion, based on a “snapshot”, but it can still supplement the study in some way. After all, taking blood and determining transcripts is also a kind of “snapshot”, since the level of these transcripts can change over time in the presence / absence of a stress factor, and, I believe, very quickly as well.
Domoic acid was detected in all four urine samples and seven of the eight fecal samples analyzed via Enzyme-linked immunosorbent assays (ELISA). Domoic acid concentrations ranged from 0.7 to 20 ng/mL in urine and 38 to 651 ng/g in feces which are consistent with those measured in urine and fecal samples from California sea lions (Zalophus californianus) with known domoic acid toxicosis. Unfortunately, the sample size of otters is 20 and only four otters were tested for domoic acid levels in their urine and eight for domoic acid levels in their feces, precluding any substantial statistical analysis. However, we have added a table with Pearson’s Correlation results and P values for relationships between gene expression and fecal domoic acid levels.
From above:
Lines 364-370 state: “However, there is no way to determine the actual domoic acid doses that these animals are experiencing based on fecal and urine domoic acid concentrations, as these samples are only a snapshot in time with no information on duration of exposure and uptake and depuration rates. The domoic acid concentrations quantified here are in the range of those measured in urine and fecal samples from California sea lions (Zalophus californianus) with known domoic acid toxicosis (Figure 2 in Lefebvre et al. [84]); however, it is not known what the total dose or duration of exposure was for any of the sea otters in our study.”
To underscore this point we have also added to the Discussion: “In addition, although in a functional system we expect a transcriptional response from certain genes upon exposure to a corresponding stressor, due to the complexities of the integrated signaling networks involved, we would not necessarily anticipate a linear relationship between fecal domoic acid level and quantity of gene transcript. With exposure there likely will be an immediate (minutes) and synchronous activation of multiple genes, followed by restriction of adaptive responses (negative feedback loops to avoid immunopathology). Simultaneously, depuration of the toxin is occurring, with the end result a very complex relationship between gene transcripts and a toxin such as domoic acid. In the future, the advent of single-cell analysis should “provide us with a comprehensive view of basic characteristics such as the presence of thresholds and signal noise in gene expression” (de Nadal et al. 2011), but at present, in follow up studies, the goal will be to refine a more general exposure/response model. Furthermore, small sample sizes precluded statistical correlation between domoic acid levels in feces and corresponding levels of gene expression.”
We have edited the concluding paragraph to highlight the limitations in our study: “The gene transcription differences observed between KBay sea otters and those from other areas suggest significant physiological perturbations in the KBay population. Although the small sample size and limited environmental data preclude definitive linkages, the patterns of altered gene transcripts and physiological systems affected, in conjunction with the detection of domoic acid in urine and fecal samples, point towards exposure to an algal toxin. However, the role of domoic acid as a cause of harmful physiological changes, and its routes, duration, and severity of exposure, warrant further investigation.”
Reviewer 2 Report
General comments:
The manuscript titled “Transcriptomics as an early warning of domoic acid toxicity” provides evidence about sea mammals may under low-level exposure to an algal toxin like domoic acid. In this study, authors try to prove the conclusion by extracting RNA from captured sea otters, running RT-PCR with several correlated genes and collecting urine or feces for toxin analyses. In a word, this research preliminarily elucidated the correlations between the algal toxins and mammal health. This article is well-written and logically clear.
1. It would be better to correct the title and some content, because the whole design did not mention about omics.
2. Has the PCR imaging system screenshot been saved? If so, please add into supplementary.

Author Response
Comments and Suggestions for Authors
General comments:
The manuscript titled “Transcriptomics as an early warning of domoic acid toxicity” provides evidence about sea mammals may under low-level exposure to an algal toxin like domoic acid. In this study, authors try to prove the conclusion by extracting RNA from captured sea otters, running RT-PCR with several correlated genes and collecting urine or feces for toxin analyses. In a word, this research preliminarily elucidated the correlations between the algal toxins and mammal health. This article is well-written and logically clear.
- It would be better to correct the title and some content, because the whole design did not mention about omics.
We have changed the title accordingly to read: “Divergent gene expression profiles in Alaskan sea otters: an indicator of chronic domoic acid exposure?”
- Has the PCR imaging system screenshot been saved? If so, please add into supplementary.
We do have the images saved. However, we would need to include at least 71 images to cover all individuals in all populations. We feel that this would be cumbersome and might actually detract from the paper’s impact.
Reviewer 3 Report
Summary
The study aimed to assess the health of sea otters lived in KBay. To do this, 13 genes were selected as biomarkers for evaluating the health of the sea otters. The expression levels of these genes in the blood of sea otters from KBay (collected in 2019), Alaska Peninsula (collected in 2009), and western Prince William Sound (1st sampling conducted from 2006 to 2008 and 2nd sampling conducted from 2010 to 2012) have been quantified using real time PCR. In addition, the amounts of domoic acid in the urine and faeces of the sea otters from KBay have been quantified and reported.
The study has reported the updated data of the sea otters lived in KBay, which is valuable to the corresponding scientific field. However, there are a few issues needed to be addressed before publication.
General comments
First of all, the title is misleading. Transcriptomics commonly refers to sequencing total RNA of a sample, followed by assembly of the raw sequencing data in order to construct the transcriptome of the sample. The study has just quantified the expressions of 13 genes in the blood of the sea otters using real time PCR. Please amend the title to avoid confusions.
Second, detailed information of the samples from the sea otters from Alaska Peninsula, and western Prince William Sound must be provided. Are they blood samples? How were they collected? When were the RNA extraction of these samples and cDNA synthesis conducted? How were the samples, the extracted RNA, and the cDNA stored before real-time PCR? Since RNAs are fragile, so it is important to provide details of the conditions of the samples and the extracted RNA samples to ensure that samples were properly stored for real time PCR analysis.
Third, the expressions of the target genes in the blood of sea otter were altered by not just the exposure of domoic acid, but also a lot of environmental abiotic or biotic stresses, like infections, and diet. Beside real-time PCR and quantification of domoic acid in the urine and faecal samples, no other experience, analysis and data has been reported to prove the direct causal relationship between the accumulation of domoic acid and the changes in expressions of those genes. The changes in the expressions of those genes can be caused by non-identified and non-reported cofounding factors in the habitat of sea otters. Moreover, the amount of domoic acid in the urine and faeces of the sea otters from Alaska Peninsula, and western Prince William Sound has no been reported in the study. What if the amounts of domoic acid in the urine and faeces were similar in all the sea otter from different locations? If it is true, then the changes of expressions of those genes were not related to the accumulation of domoic acid. Therefore, no enough evidence has been provided to support the accumulation of domoic acid is responsible to the changes in the expressions of those genes.
Based on the 3rd comment, 3 amendments are suggested:
1 1. The title must be amended. The current title “Transcriptomics as an early warning of domoic acid toxicity” is misleading. When the audiences read the title, it seems that the study has already proved the casual relationship between the accumulation of domoic acid and the changes of the expressions of those genes in the blood of the sea otters. In fact, to me, the casual relationship has not been proved.
2 2. Please provide some information about the environmental abiotic and biotic factors in the sea otter habitat that are known to alter the expressions of those target genes in the blood. So, the audiences can be informed that there are some other cofounding factors.
3 3. I do understand that the difficulties on collecting field samples. The research team may not have urine and faecal samples from the sea otters from Alaska Peninsula, and western Prince William Sound for comparison. However, the research team can test whether the amounts of domoic acid in urine and faeces have linear correlations with the expressions of those genes in blood. If the expressions of the target genes showed some significant linear correlations with the amount of domoic acid in urine and faeces, this will give grounds for the idea that the accumulation of domoic acid may affect the expressions of genes in blood of sea otter. If so, it will be worth to further investigate the casual relationship between the exposure to domoic acid and the expressions of the genes in blood.
Specific comments
Line 201: any post-hoc analysis conducted after the anova test?
Line 215 – 229, please provide the p values for the anova tests for the ct values from different samples in either Table 2 or Figure 1

Author Response
Comments and Suggestions for Authors
Summary
The study aimed to assess the health of sea otters lived in KBay. To do this, 13 genes were selected as biomarkers for evaluating the health of the sea otters. The expression levels of these genes in the blood of sea otters from KBay (collected in 2019), Alaska Peninsula (collected in 2009), and western Prince William Sound (1st sampling conducted from 2006 to 2008 and 2nd sampling conducted from 2010 to 2012) have been quantified using real time PCR. In addition, the amounts of domoic acid in the urine and faeces of the sea otters from KBay have been quantified and reported.
The study has reported the updated data of the sea otters lived in KBay, which is valuable to the corresponding scientific field. However, there are a few issues needed to be addressed before publication.
General comments
First of all, the title is misleading. Transcriptomics commonly refers to sequencing total RNA of a sample, followed by assembly of the raw sequencing data in order to construct the transcriptome of the sample. The study has just quantified the expressions of 13 genes in the blood of the sea otters using real time PCR. Please amend the title to avoid confusions.
We have changed the title accordingly to read: “Divergent gene expression profiles in Alaskan sea otters: an indicator of chronic domoic acid exposure?”
Second, detailed information of the samples from the sea otters from Alaska Peninsula, and western Prince William Sound must be provided. Are they blood samples? How were they collected? When were the RNA extraction of these samples and cDNA synthesis conducted? How were the samples, the extracted RNA, and the cDNA stored before real-time PCR? Since RNAs are fragile, so it is important to provide details of the conditions of the samples and the extracted RNA samples to ensure that samples were properly stored for real time PCR analysis.
We have added the following sentence at the end of section 2.1: “Methods for otter capture and sample processing and analysis were identical among all populations and are detailed below in sections 2.2 – 2.5.”
Third, the expressions of the target genes in the blood of sea otter were altered by not just the exposure of domoic acid, but also a lot of environmental abiotic or biotic stresses, like infections, and diet. Beside real-time PCR and quantification of domoic acid in the urine and faecal samples, no other experience, analysis and data has been reported to prove the direct causal relationship between the accumulation of domoic acid and the changes in expressions of those genes. The changes in the expressions of those genes can be caused by non-identified and non-reported cofounding factors in the habitat of sea otters. Moreover, the amount of domoic acid in the urine and faeces of the sea otters from Alaska Peninsula, and western Prince William Sound has no been reported in the study. What if the amounts of domoic acid in the urine and faeces were similar in all the sea otter from different locations? If it is true, then the changes of expressions of those genes were not related to the accumulation of domoic acid. Therefore, no enough evidence has been provided to support the accumulation of domoic acid is responsible to the changes in the expressions of those genes.
Please see responses to 1, 2, and 3 below.
Based on the 3rd comment, 3 amendments are suggested:
- The title must be amended. The current title “Transcriptomics as an early warning of domoic acid toxicity” is misleading. When the audiences read the title, it seems that the study has already proved the casual relationship between the accumulation of domoic acid and the changes of the expressions of those genes in the blood of the sea otters. In fact, to me, the casual relationship has not been proved.
We have changed the title accordingly to read: “Divergent gene expression profiles in Alaskan sea otters: an indicator of chronic domoic acid exposure?”
- Please provide some information about the environmental abiotic and biotic factors in the sea otter habitat that are known to alter the expressions of those target genes in the blood. So, the audiences can be informed that there are some other cofounding factors.
In Table 1 we have included factors that can influence the expression levels of each gene in our panel. Additionally, we have revised our concluding paragraph to specifically point this out: “The gene expression differences observed between KBay sea otters and those from other areas suggest significant physiological perturbations in the KBay population. The expression levels of these five genes can be modified by more than one environmental stressor (Table 1). This, in combination with the lack of extensive environmental data, precludes us from making definitive linkages between domoic acid and gene expression patters. However, the patterns of altered gene transcripts and physiological systems affected, in conjunction with the detection of domoic acid in urine and fecal samples, point towards exposure to an algal toxin. However, the role of domoic acid as a cause of harmful physiological changes, and its routes, duration, and severity of exposure, warrant further investigation.”
- I do understand that the difficulties on collecting field samples. The research team may not have urine and faecal samples from the sea otters from Alaska Peninsula, and western Prince William Sound for comparison. However, the research team can test whether the amounts of domoic acid in urine and faeces have linear correlations with the expressions of those genes in blood. If the expressions of the target genes showed some significant linear correlations with the amount of domoic acid in urine and faeces, this will give grounds for the idea that the accumulation of domoic acid may affect the expressions of genes in blood of sea otter. If so, it will be worth to further investigate the casual relationship between the exposure to domoic acid and the expressions of the genes in blood.
Unfortunately, the sample size of otters is 20 and only four otters were tested for domoic acid levels in their urine and eight for domoic acid levels in their feces, precluding any substantial statistical analysis. However, domoic acid was detected in all four urine samples and seven of the eight fecal samples analyzed via Enzyme-linked immunosorbent assays (ELISA). Domoic acid concentrations ranged from 0.7 to 20 ng/mL in urine and 38 to 651 ng/g in feces which are consistent with those measured in urine and fecal samples from California sea lions (Zalophus californianus) with known domoic acid toxicosis.
From above:
Lines 364-370 state: “However, there is no way to determine the actual domoic acid doses that these animals are experiencing based on fecal and urine domoic acid concentrations, as these samples are only a snapshot in time with no information on duration of exposure and uptake and depuration rates. The domoic acid concentrations quantified here are in the range of those measured in urine and fecal samples from California sea lions (Zalophus californianus) with known domoic acid toxicosis (Figure 2 in Lefebvre et al. [84]); however, it is not known what the total dose or duration of exposure was for any of the sea otters in our study.”
To underscore this point we have also added to the discussion: “In addition, although in a functional system we expect a transcriptional response from certain genes upon exposure to a corresponding stressor, due to the complexities of the integrated signaling networks involved, we would not anticipate a linear relationship between fecal domoic acid level and quantity of gene transcript. With exposure there will be an immediate (minutes) and synchronous activation of multiple genes, followed by restriction of adaptive responses (negative feedback loops to avoid immunopathology). Simultaneously, depuration of the toxin is occurring, with the end result a very complex relationship between gene transcripts and a toxin such as domoic acid. In the future, the advent of single-cell analysis should “provide us with a comprehensive view of basic characteristics such as the presence of thresholds and signal noise in gene expression” (de Nadal et al. 2011), but at present the more general exposure/response model must suffice. Furthermore, small sample sizes precluded statistical correlation between domoic acid levels in feces and corresponding levels of gene expression.”
We have edited the concluding paragraph to highlight the limitations in our study: “The gene expression differences observed between KBay sea otters and those from other areas suggest significant physiological perturbations in the KBay population. The expression levels of these five genes can be modified by more than one environmental stressor (Table 1). This, in combination with the lack of extensive environmental data, precludes us from making definitive linkages between domoic acid and gene expression patters. However, the patterns of altered gene transcripts and physiological systems affected, in conjunction with the detection of domoic acid in urine and fecal samples, point towards exposure to an algal toxin. Subsequently, the role of domoic acid as a cause of harmful physiological changes, and its routes, duration, and severity of exposure, warrant further investigation.”
Specific comments
Line 201: any post-hoc analysis conducted after the anova test?
Yes, we stated: “Differences in means were calculated using Kruskal-Wallis non-parametric rank ANOVA test which included a Dunn’s Test for multiple comparisons (NCSS, Statistical and Power Analysis Software, UT, USA).”
Line 215 – 229, please provide the p values for the anova tests for the ct values from different samples in either Table 2 or Figure 1
We have provided P values for each ANOVA test as well as letters signifying group differences according to the Dunn’s Test.
Round 2
Reviewer 1 Report
Dear colleagues,
Thank you for the thorough discussion of the issues that have arisen and for making appropriate comments to the manuscript.
Reviewer 3 Report
I am satisfied with the revised manuscript, expect for the table 2. I can only read the first 5 columns of the table, but it seems that the table should have more than 5 columns. So please ensure audiences can read all the columns of table 2. After this amendment, the manuscript is ready for publication.